

# Predicting the effect of variants on splicing using Convolutional Neural Networks

Thanyathorn Thanapattheerakul[1], Worrawat Engchuan[2,3] and Jonathan H. Chan[1,4]

[1] School of Information Technology, King Mongkut's University of Technology Thonburi, Bangkok, Thailand
[2] Genetics and Genome Biology, The Hospital for Sick Children, Toronto, Ontario, Canada
[3] The Centre for Applied Genomics, The Hospital of Sick Children, Toronto, Ontario, Canada
[4] IC2-DLab, School of Information Technology, King Mongkut's University of Technology Thonburi, Bangkok, Thailand

Corresponding author
Jonathan H. Chan,
jonathan@sit.kmutt.ac.th

## ABSTRACT

Mutations that cause an error in the splicing of a messenger RNA (mRNA) can lead to diseases in humans. Various computational models have been developed to recognize the sequence pattern of the splice sites. In recent studies, Convolutional Neural Network (CNN) architectures were shown to outperform other existing models in predicting the splice sites. However, an insufficient effort has been put into extending the CNN model to predict the effect of the genomic variants on the splicing of mRNAs. This study proposes a framework to elaborate on the utility of CNNs to assess the effect of splice variants on the identification of potential disease-causing variants that disrupt the RNA splicing process. Five models, including three CNN-based and two non-CNN machine learning based, were trained and compared using two existing splice site datasets, Genome Wide Human splice sites (GWH) and a dataset provided at the Deep Learning and Artificial Intelligence winter school 2018 (DLAI). The donor sites were also used to test on the HSplice tool to evaluate the predictive models. To improve the effectiveness of predictive models, two datasets were combined. The CNN model with four convolutional layers showed the best splice site prediction performance with an AUPRC of 93.4% and 88.8% for donor and acceptor sites, respectively. The effects of variants on splicing were estimated by applying the best model on variant data from the ClinVar database. Based on the estimation, the framework could effectively differentiate pathogenic variants from the benign variants ($p = 5.9 \times 10^{-7}$). These promising results support that the proposed framework could be applied in future genetic studies to identify disease causing loci involving the splicing mechanism. The datasets and Python scripts used in this study are available on the GitHub repository at https://github.com/smiile8888/rna-splice-sites-recognition.

# INTRODUCTION

RNA splicing, a process exclusive to eukaryotic cells, is a post-transcriptional modification of a protein-coding messenger RNA (mRNA). This process is carried out by a complex of

small nuclear RNA (snRNA) and protein, known as a spliceosome, which binds to the splice site on a pre-mRNA to fold, clip and rejoin the pre-mRNA. The intronic sequence is then eliminated, and the remaining exonic sequences are joined. The whole process is referred to as RNA splicing (*Faustino & Cooper, 2003*). A single pre-mRNA can be encoded to multiple proteins by a mechanism called alternative splicing, where the pre-mRNA is modified by incorporating different sets of the exons. This mechanism allows the eukaryotic genome to store more information economically but requires precise regulation. In humans, errors in splicing or mis-splicing have been shown to underlie many diseases, including heart disease, dementia, and autism spectrum disorder (ASD) (*Scotti & Swanson, 2016*). Specifically, a mis-splicing of LDB3 regulated by RBM20 leads to the development of heart disease (*Zhu et al., 2017*; *Rexiati, Sun & Guo, 2018*), and mutations in MAPT can cause an increase in the splicing of its exon that leads to frontotemporal dementia with Parkinsonism (*Buée et al., 2000*). More complex conditions like schizophrenia and autism spectrum disorder (ASD) have also been linked to mis-splicing caused by single nucleotide variations (SNVs) (*Reble, Dineen & Barr, 2018*).

Many studies have used position-weight-matrix (PWM) to recognize the sequence pattern of DNA/RNA binding sites, including splice sites (*Stormo, 2000*). Knowing the sequence pattern and binding specificity makes it possible to assess the effect of SNVs on the binding affinity at the sequence level (*Desmet et al., 2010*). Although PWM is powerful and easy to interpret, it can capture only a simple sequence pattern; for more complex sequence patterns, a more advanced method is required. Over the past few years, whole genome sequencing data have been increasingly deposited in public databases due to the advances in sequencing technology and lowered costs (*Stephens et al., 2015*; *Lek et al , 2016*). The enormous amount of publicly available data has allowed more analytic methods to emerge, with some being improved versions of existing techniques and some being novelties.

Recently, machine learning (ML) and deep learning (DL) have been applied to solve problems in many fields with astonishing results, especially for applications in the computer vision field (*LeCun, Bengio & Hinton, 2015*). It has also been adopted in biomedical research (*Wainberg et al., 2018*). Support Vector Machines (SVMs) and Random Forest (RF) are the most popular of traditional ML techniques. *Sonnenburg et al. (2007)* proposed the SVM with a weighted degree kernel to recognize splice sites. Various SVM- and RF-based tools, for example, HSplice (*Meher et al., 2016*) and MaLDoSS (*Meher, Sahu & Rao, 2016*), have been made available in the public domain for the prediction of donor splice sites in many species, including *Homo sapiens, Bos taurus, Danio rerio,* and *Caenorhabditis elegans.* Convolutional neural networks (CNNs) have been leveraged to identify the sequence motifs of the binding sites in the human genome. DeepSEA and DeepBind are CNN-based algorithms designed to capture the sequence specificity of DNA/RNA binding proteins and assess the impact of SNVs on the binding sites (*Alipanahi et al., 2015*; *Zhou & Troyanskaya, 2015*). SpliceRover, a CNN-based tool for splice site prediction, demonstrated improved performance compared to conventional SVMs (i.e., linear SVM and SVM with a weighted degree kernel) and the deep belief network (i.e., restricted Boltzmann machine (RBM)) on different splice site datasets (*Zuallaert et al., 2018*). Recurrent Neural Network (RNN)

and its variants, such as Long Short-Term Memory (LSTM), have also been adopted to solve classification problems on DNA and RNA sequences. Quang and Xie proposed a hybrid convolutional and recurrent deep neural network to predict the transcription factor binding sites. They made use of a CNN to capture the features in the sequences, while a RNN learned the relationship among those features. Their comparison results with DeepSEA showed that using a hybrid model improves the performance by more than 50% over using CNN alone (*Quang & Xie, 2016*).

Although deep learning techniques have shown superior performance over other methods, no framework has been built to extend the model to estimate the effect of genomic variants located near the splice sites. Therefore, a framework is proposed here to predict the effects of such variants on splicing events, in the hopes of identifying the variants causing disease through disruption of the splicing mechanism.

## Framework

The framework covers two parts: (1) training a model for splice site prediction; and (2) estimating the effect of variants on splicing. In particular, the pipeline used in this study involved the comparison of deep learning and traditional machine learning models to recognize sequence patterns of the splice sites on two datasets. Then the model with the best performance in distinguishing the actual splice sites from the negative sequences was used to estimate the effect of the point mutation on the splice site (referred here as 'splice variant' or 'variant'). The overall workflow of this framework is shown in Fig. 1.

The model preparation starts from data gathering and preprocessing, followed by modeling and validation of the model with unseen data. Data gathering and preprocessing are essential steps because the quality and quantity of the data directly affect the performance of the model. The preprocessing step involves the conversion of raw data into a compatible format as the input of the model. The model can be based on traditional machine learning algorithms, deep learning techniques, or other algorithms that give a probabilistic prediction. The probabilistic prediction is required to calculate a score for the splice variants.

The second part is to estimate the effect of the splice variant by using the pre-trained model from the previous step. For each variant, a reference (major allele) and an alternative (minor allele) sequence of splice sites where the variant is located was obtained, then the model was applied on the obtained sequences to give a probability of being a splice site. If the variant affected the splice site in some way, it was inferred that the probabilistic prediction on the reference sequence was higher than the one with an alternative variant. The variants affecting the splice sites could result in mis-splicing and be disease-causing if an important gene is disrupted. A score for each pair of sequences was calculated by taking the difference between the probabilistic prediction of reference and alternative sequences.

## MATERIALS & METHODS

### Dataset

As mentioned earlier, two datasets were used in this work. The first one is called Genome-Wide Human splice sites (GWH), which was obtained from a 2007 paper by *Sonnenburg et*
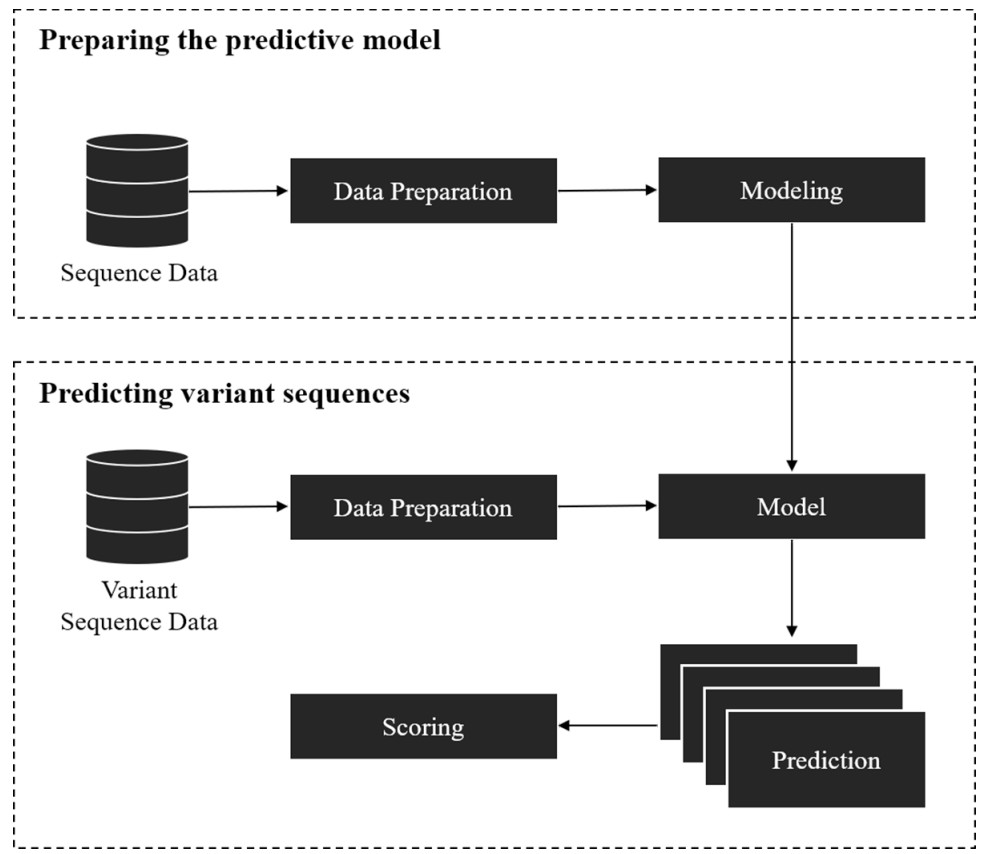

**Figure 1** **Overall workflow of the proposed variant scoring framework.**

*al. (2007).* The splice site data were retrieved by processing the Expressed-Sequence Tags (ESTs) and cDNA data available in commonly used databases (*Sonnenburg et al., 2007*). The GWH data is an imbalanced case-control dataset where cases or positive sequences are confirmed sequences of being spliced sites, and controls or negative sequences are other sequences with spliced-site core dinucleotide at the center position of the sequences. For donor sites, there are 1,484,844 sequences of negative data, while 80,515 sequences are positive data. Similarly, 1,374,182 sequences of acceptor sites are negative data, while 79,250 sequences are positive data. Each sequence has a length of 398 nucleotides (nt) with core dinucleotides, GT for donor sites and AG for acceptor sites, in the middle of the sequence. Specifically, the dimer of donor sites is at the position of 201 and 202, while 198 and 199 are the position of the dimer in acceptor sites.

The second dataset is called DLAI. It was a dataset provided in the competition track of the First Deep Learning and Artificial Intelligence Winter School (DLAI1) (https://deeplearningandaiwinterschool.github.io/dlai1.html). The dataset was prepared by obtaining transcript data from the curated RefSeq database (hg19). Many of the splicing signals are found within ±50 nt window from the splice site based on the average absolute weighted contribution score (wcs) (*Zuallaert et al., 2018*); however, the majority is mainly

within ±20 nt window of the splice sites. Thus, sequences of ±20 nt window from the splice site were extracted. This was done to reduce noise from extending the window for too far and also to speed up the training process. The negative sequences were defined as non-splice site sequences (40 nt) containing core dinucleotides. The negative donor sequences have a GT dimer at position 20 and 21, while the acceptor negative sequences have an AG dimer at position 19 and 20. Unlike the GWH dataset, this is a balanced dataset which contains 223,143 sequences of the donor, and 220,034 sequences of the acceptor. Even though the preparation of the two datasets was different, overlapping levels of positive sequences between the two were very high, as 74% of donor and 75% of acceptor sites of GWH data were found in the DLAI data. On the other hand, the negative sequences rarely overlapped between both datasets, as only 1% of donor and 2% of acceptor sites from the GWH data were found in the DLAI data.

The splice variant data was gathered from the ClinVar database. The ClinVar is a clinical variant database, which contains both copy number variations (CNVs) and single nucleotide variations (SNVs). All the variants were classified into different groups based on their pathogenicity level by manual validation and/or computational method following the American College of Medical Genetics and Genomics (ACMG) guidelines for Mendelian disorder variant interpretation (*Landrum et al., 2014*). The SNVs belonging to benign, likely benign, likely pathogenic, and pathogenic groups were obtained to validate the model in the second part of the framework. The pathogenic variants are genomic variants with evidence reported that they cause a disease. To evaluate the performance of the model, the splicing effect scores estimated by the model SNVs were compared between pathogenic (pathogenic + likely pathogenic; $n = 801$ and $n = 356$, for donor and acceptor, respectively) and benign (benign + likely benign; $n = 11,200$ and $n = 10,944$, for donor and acceptor, respectively) variants. Only variants located within 20 base pairs (bp) from splice sites were obtained along with their respective splice site sequences (40 bp of length with splice junction at the center of the sequence).

## Methodology

This section describes a case study applied to the proposed framework, including the process of building predictive models and predicting the effect of splice variants.

### Data Preprocessing

The predictive models used in the study are based on deep learning and traditional machine learning techniques. The input of these models requires a numerical matrix representing pixels in a black and white or a grayscale image. Thus, the input sequences have to be transformed into a matrix format. A DNA sequence is a string composed of four letters: A, C, G, and T. These correspond to the vectors $[1, 0, 0, 0]$, $[0, 1, 0, 0]$, $[0, 0, 1, 0]$, and $[0, 0, 0, 1]$, respectively. Therefore, each sequence is represented as a $N \times 4$ matrix, where $N$ is the length of the input DNA sequence, and 4 is the number of different nucleotides. Figure 2 illustrates an input DNA sequence and the corresponding matrix. This transformation technique is called one-hot encoding.
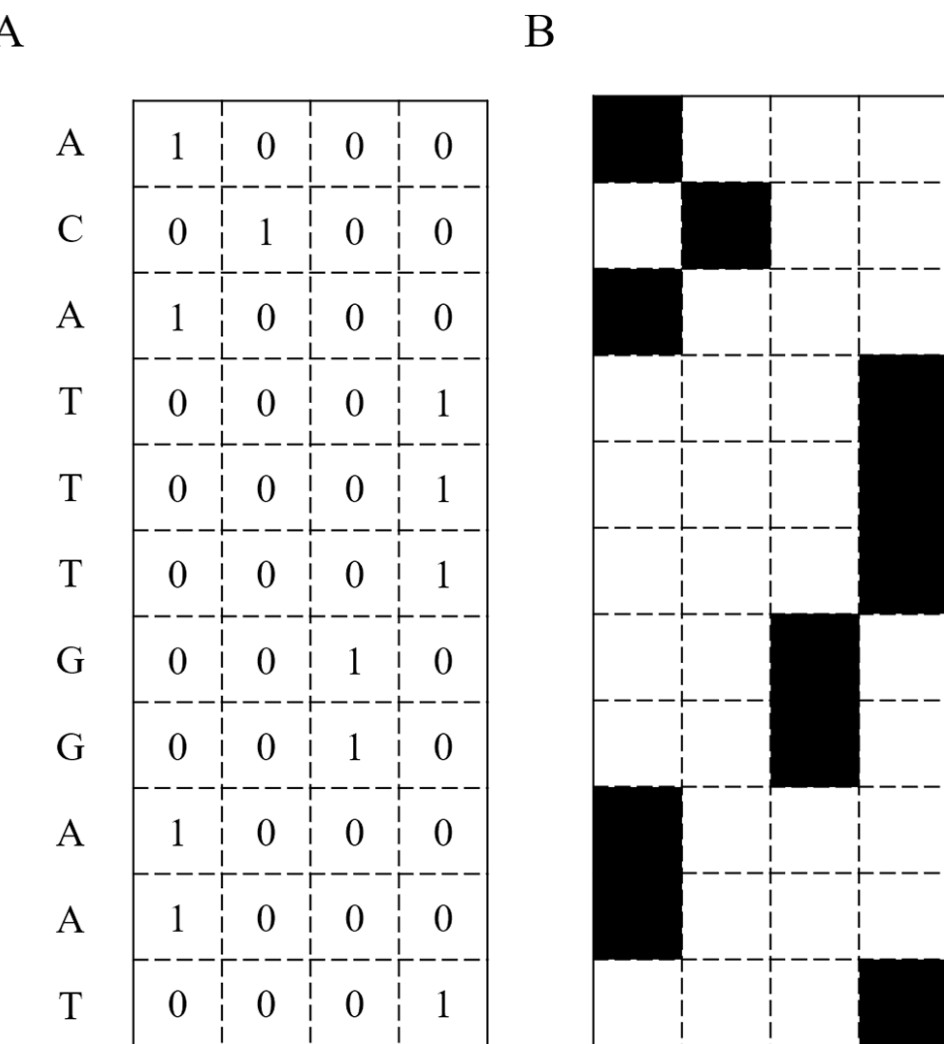

**Figure 2** **DNA Sequence converted into a binary matrix.** (A) A DNA sequence transformed into a binary matrix. (B) The DNA sequence represented as an image.

### Modeling

To recognize splice sites, any model that can give a probabilistic prediction can be applied. In this study, a comparison of deep learning, to traditional machine learning algorithms—convolutional neural networks (CNNs), and a hybrid CNN with LSTM, to support vector machine (SVM) and random forest (RF)—is provided.

The NNs are designed for representation of high-level abstraction in the data. Typically, NNs consist of an input layer, hidden layers, and an output layer, each consisting of a number of neurons or nodes. Each neuron is a processing unit with different parameters or weights. The input layer propagates the data through the network, yielding intermediate results using an activation function at each hidden layer. The output layer results in a final prediction. For more complex models, a nonlinear function, e.g., Rectified Linear Unit (ReLU) (*Nair & Hinton, 2010*) and softmax, is applied to activate neurons in each layer

so that they are able to represent a non linear relationship. Over the past few years, the ReLU has been applied to activate neurons in hidden layers. It is the simplest nonlinear activation function, which outputs 0 if the input is less than or equal 0, and outputs raw output, otherwise. In classification problems, softmax is commonly used as an activation function in the last layer to generate a probability of each class. In supervised learning, NNs learn from the annotated training data by adjusting the weights based on a loss function. The loss function represents the difference between the predictions from the network and the annotated labels. Besides traditional NNs, convolutional neural networks (CNNs) have been proposed and shown to outperform traditional NNs in image recognition, including classifying images, detecting objects, and recognizing faces. CNN is a variation of NNs that consists of at least one convolutional layer in the NNs. The convolutional layer has filters that slide over the sequence and detect patterns. Here, weights are stored within a filter to be shared over different positions. Typically, the convolutional layer is followed by a pooling layer, which helps to reduce dimensionality and map features independently. The most common approach used in the pooling layer is Max-pooling, which uses a maximum value representing the area of the specified filter. Overfitting is the main issue when NNs have many layers, resulting in high performance in training data but poor performance on unseen data. Neuron dropout is a common technique used to avoid the over-fitting issue by randomly deactivating some neurons from the network which helps to reduces independent learning among neurons (*Srivastava et al., 2014*).

Another variation of NNs is recurrent neural networks (RNNs). The RNN has an internal loop to maintain a cell state of extracted information. Instead of processing the whole sequence in a single step, it processes the sequence by iterating through the sequence elements and allowing information relative to what it has processed to persist. The information stored in the internal state allows the network to exhibit dynamic temporal or spatial behavior (*Schuster & Paliwal, 1997*). Although the RNN should theoretically be able to relate previous information to the present extracted information, in practice, as the sequence grows, it becomes unable to learn to connect the information. Connections between past and present information are called "long-term dependencies." Long Short-Term Memory networks (LSTMs) have been introduced to solve this problem (*Bengio, Simard & Frasconi, 1994*; *Hochreiter & Schmidhuber, 1997*). The LSTM is the variant of RNN that is capable of learning long-term dependencies. It is able to add or remove information to the internal cell states. The amounts of information to be added or removed are carefully regulated by structures called gates. A bi-directional LSTM is a variant of standard LSTM that combines two LSTMs, where each takes a sequence in a different direction; for example, in sequential data, one moves from left to right, and the other moves right to left.

Machine learning techniques like SVMs and RFs are known to be excellent for classification tasks. In the bioinformatics field, SVMs and RFs have been applied to predict splice sites. They both gave a promising performance as reported in papers by Sonnenburg, Lee and Yoon (*Sonnenburg et al., 2007*; *Lee & Yoon, 2015*).

In this case study, five models were used: two CNN-based models, the CNN with bidirectional LSTM model, the SVM model, and the RF model, to distinguish the positive

and negative sequences. The CNN-based models include the model for the sequence length of 40 nucleotides from SpliceRover (*Zuallaert et al., 2018*) and the other one from our preliminary study (*Thanapattheerakul et al., 2018*). These models are called CNN_3 and CNN_4 as the number represents the number of convolutional layers in the model. The hybrid CNN with bidirectional LSTM, the architecture of which was derived from DanQ (*Quang & Xie, 2016*), is called CNN_LSTM. The bi-directional LSTM is integrated with a CNN at the last layer before being connected to the fully connected layer. All architectures and hyperparameters of CNN and CNN with LSTM models are shown in Tables 1 and 2, respectively. For SVM and RF, the recommended default hyperparameters were used.

### Model Comparison

To evaluate the different models, 5-fold cross-validation was performed on the DLAI data. The DLAI was used because it is a balanced dataset containing the most up-to-date sequences. When comparing to the other datasets, it is the smallest one, but it was sufficient to train models, so the training time was reduced. The 5-fold cross-validation was done by the traditional cross-validation approach that resamples the data into five portions. In each iteration, four portions are used as a training data to fit the model, while the rest is used as a hold-out to evaluate the model. The model is then discarded and re-declared in the next iteration. As an additional comparison of the developed CNN models to the traditional machine learning approach, an existing web-based tool termed HSplice (*Meher et al., 2016*) was used. This tool is available for prediction of human donor splice sites only, and it is available at http://cabgrid.res.in:8080/HSplice. Prior to using the HSplice tool, the donor sites needed to be shortened from 40 bp to 15 bp ($-8$ to $+7$ from splice junction). The output of the tool was the prediction probability of the given sequence being a splice site. For a direct comparison with this work, the prediction probability was used to calculate the area under precision and recall curve (AUPRC) and area under the receiver operating characteristic (AUROC). To obtain precision and recall, the same procedure was used to convert the probability of each sequence to a binary class output, using a threshold of 0.5; if the probability was greater than 0.5, it was classified as splice site, otherwise it was a nonsplice site.

### Testing effect of imbalanced data

To maximize statistical power, the two datasets, GWH and DLAI, were combined to increase the number of positive and negative sequences. The sequences from GWH were trimmed down from 398 nt to 40 nt to match with the ones from DLAI. Although the combined dataset was much more comprehensive, it became imbalanced. For donor sites, there were 230,208 positive sequences and 1,669,934 negative sequences. For acceptor sites, there were 226,436 positive sequences and 1,558,077 negative sequences. The combined dataset was used to test the effect of imbalanced data in order to utilize the model performance. Since the positive data is in the minority, 80% of the positive data was randomly selected. Also, subsets of negative data were then picked to construct different datasets where the ratio of positive to negative data was restricted at 1:1, 1:3, 1:5, and 1:7. The CNN_3 and CNN_4 models were then validated by performing 5-fold cross-validation using these subsets of positive and negative sequences. Furthermore, after training on the different subsets, each

**Table 1 CNN-based architecture details.** The details of the architecture of CNN_3, CNN_4, and CNN_LSTM are described. Both CNN_3 and CNN_4 are CNN-based architecture, but they are different in the number of layers and the filter in each layer. The CNN_LSTM is a hybrid CNN with bi-directional LSTM.

| Name | Architectures | |
|---|---|---|
| | **Layers** | **Details** |
| CNN_3 | conv2D layer 1 | 70 filters of size (9,4) |
| | dropout layer 1 | $p = 0.2$ |
| | conv2D layer 2 | 100 filters of size (7,1) |
| | maxpool layer 1 | pool size (2,1) |
| | dropout layer 2 | $p = 0.2$ |
| | conv2D layer 3 | 150 filters of size (7,1) |
| | maxpool layer 2 | pool size (2,1) |
| | dropout layer 3 | $p = 0.2$ |
| | dense layer 1 | 512 neurons |
| | dropout layer 4 | $p = 0.2$ |
| | softmax layer | 2 outputs |
| CNN_4 | conv2D layer 1 | 70 filters of size (3,4) |
| | dropout layer 1 | $p = 0.2$ |
| | conv2D layer 2 | 100 filters of size (3,1) |
| | dropout layer 2 | $p = 0.2$ |
| | conv2D layer 3 | 100 filters of size (3,1) |
| | maxpool layer 1 | pool size (2,1) |
| | dropout layer 3 | $p = 0.2$ |
| | conv2D layer 4 | 200 filters of size (3,1) |
| | maxpool layer 2 | pool size (2,1) |
| | dropout layer 4 | $p = 0.2$ |
| | dense layer 1 | 512 neurons |
| | dropout layer 5 | $p = 0.2$ |
| | softmax layer | 2 outputs |
| CNN_LSTM | conv1D layer 1 | 320 filters of length 26 |
| | maxpool layer 1 | pool size (13) |
| | dropout layer 1 | $p = 0.2$ |
| | bidirectional LSTM layer 1 | 320 output dimension |
| | dropout layer 2 | $p = 0.5$ |
| | Dense layer 1 | 925 neurons |
| | softmax layer | 2 outputs |

model was tested on the remaining 20% of positive data and the rest of the negative data. As an additional comparison, 20% of the donor site test set was randomly selected to test on the CNN models and HSplice tool.

### Testing effect of window sizes

Another factor that could affect the model performance is window size or input sequence length. To compare the effect of window size, the CNN_4 model was performed on the GWH dataset with five different sequence lengths: 40 nt, 80 nt, 160 nt, 240 nt, and 398 nt.
**Table 2  CNN-based hyperparameter details.** The hyperparameters set applied for CNN_3, CNN_4, and CNN_LSTM.

| Optimizer | Loss function | Epoch | Batch size | Start learning rate | # Steps per learning rate decay | Learning rate decay scheduling |
|---|---|---|---|---|---|---|
| SGD with Nesterov momentum 0.9 | Categorical cross-entropy | 30 | 64 | 0.05 | 5 | Yes |

The subset of the GWH dataset with a 1:7 ratio of positive to negative sequencing from the previous step was used in 5-fold cross-validation.

### Predicting variant effect

To predict the effects of variants, the CNN_4 model was used on the combined dataset with a 1:7 ratio of positive to negative sequences. This dataset was used since it contains the most up-to-date data (DLAI dataset) and a bigger negative set. This model was tested on the splice variants from the ClinVar database. The effects of variants on splicing were estimated by making a probabilistic prediction of whether splicing would occur on the sequence with the presence and the absence of an alternative allele. Then, the variant was scored by taking the difference between the two predictions using the formula below as suggested by a previous study of the effect of variants on transcription factor binding site (*Zhou & Troyanskaya, 2015*):

$$Score(m_i) = \log 10 \left( \frac{PM(ref)}{PM(alt)} \right)$$

where $Score(m_i)$ is a score of variant or mutation, $PM(ref)$ is a probability of a reference sequence being a splice site (*reference sequence:* sequence with the absence of an alternative allele), and $PM(alt)$ is the probability of an alternative sequence being a splice site (*alternative sequence:* sequence with the presence of an alternative allele).

The splice variant effect prediction was obtained from the dbscSNV database, used to compare with the prediction from the proposed method. The dbscSNV is a comprehensive database that stores splicing effect score (ada_score) of human SNVs located in the splicing regions, i.e., 3 to +8 from donor sites and $-12$ to $+2$ from acceptor sites (*Jian, Boerwinkle & Liu, 2014*). The ada_score is a prediction score (in the range of 0 to 1) of variants causing splicing disruption and leading to disease computed using AdaBoost model (*Jian, Boerwinkle & Liu, 2014*). It was compared to the score predicted by the proposed method based on how well they distinguished benign and pathogenic variants.

### Computational setup

All experiments of CNN- and RF-based models were conducted on Google Colaboratory (https://colab.research.google.com/). The backend was run with Python 3 and a GPU hardware accelerator. Due to the time limitation (12 h per session for the freely available resource) on Google Colaboratory, the SVM-based model was run on a local laptop without GPU (Intel Core i5-3230M CPU @ 2.60 GHz, x64-based Windows OS, 8 GB of RAM, 256 GB SSD). Pandas (*McKinney, 2010*) and NumPy (*Oliphant, 2006*; *VanDer Walt, Colbert & Varoquaux, 2011*) were used in the processes of data preparation and representation. The ggplot2 R package (*Wickham, 2009*) and matplotlib (*Hunter, 2007*) were used for

**Table 3  The performance of five predictive models.** The average AUPRC, precision, recall, AUROC, and the average training time of the five predictive models from 5-fold cross-validation are described. For the donor sites, the HSplice tool was used as a benchmark.

| Site | Model | AUPRC | | Precision | | Recall | | AUROC | | Runtime (Colab) |
|------|-------|-------|----|-----------|----|--------|----|-------|----|------------------|
| | | mean | SD | mean | SD | mean | SD | mean | SD | |
| Donor | CNN_3 | **0.986** | 0.0005 | 0.936 | 0.0013 | 0.979 | 0.0009 | **0.989** | 0.0003 | 12 m |
| | CNN_4 | **0.986** | 0.0002 | 0.930 | 0.0015 | **0.982** | 0.0010 | **0.989** | 0.0001 | 12 m |
| | CNN_LSTM | 0.983 | 0.0004 | 0.932 | 0.0003 | 0.975 | 0.0013 | 0.986 | 0.0002 | 25 m |
| | SVM | 0.923 | 0.0007 | 0.937 | 0.0007 | 0.968 | 0.0012 | 0.952 | 0.0006 | 2 hr[a] |
| | RF | 0.913 | 0.0004 | **0.939** | 0.0006 | 0.942 | 0.0007 | 0.940 | 0.0002 | 11 s |
| | HSplice | 0.968 | | 0.928 | | 0.936 | | 0.975 | | N/A |
| Acceptor | CNN_3 | **0.979** | 0.0003 | 0.910 | 0.0028 | 0.968 | 0.0027 | **0.982** | 0.0004 | 12 m |
| | CNN_4 | **0.979** | 0.0008 | 0.905 | 0.0030 | **0.973** | 0.0012 | **0.982** | 0.0006 | 12 m |
| | CNN_LSTM | 0.975 | 0.0008 | 0.914 | 0.0020 | 0.960 | 0.0013 | 0.979 | 0.0006 | 25 m |
| | SVM | 0.893 | 0.0017 | **0.915** | 0.0018 | 0.948 | 0.0013 | 0.930 | 0.0013 | 2.30 hr[a] |
| | RF | 0.866 | 0.0009 | 0.910 | 0.0011 | 0.893 | 0.0020 | 0.902 | 0.0010 | 11 s |

**Notes.**
[a]The SVM-based model was run on a local laptop without GPU (Intel Core i5-3230M CPU 2.60 GHz, x64-based Windows OS, 8 GB of RAM, 256 GB SSD).

Bold styling emphasizes the highest values regarding the evaluation metrics used in the study.

data and result visualization, including graphic generation. Keras (https://keras.io/) with TensorFlow was used for model construction, training and testing.

## RESULTS

### Comparison between deep learning and traditional machine learning approaches

The first set of results is a comparison between CNNs, CNN_LSTM, SVM, and RF in splice site prediction performance assessed by 5-fold cross-validation on the DLAI dataset only, which is a balanced dataset. Moreover, for donor sites, HSplice, a donor site prediction tool, was used as a benchmark in the comparison. Here, several evaluation metrics are reported, including precision, recall, AUPRC, and AUROC. Table 3 shows the performance on the 5-fold cross-validation and the average training time, which was also obtained from 5-fold cross-validation of all models. The CNN-based models performed significantly better than SVM and RF models regarding the AUPRC (One-sided Welch's $t$-test $p = 6 \times 10^{-12}$ and $p = 4.8 \times 10^{-15}$, for SVM and RF, respectively). Among the CNN-based models, CNN_3 and CNN_4 outperformed the CNN_LSTM (One-sided Welch's $t$-test $p = 1.4 \times 10^{-5}$), while no significant difference in performance between the CNN_3 and CNN_4 models was found (One-sided Welch's $t$-test $p = 0.54$). Besides the superior performance, the training times of CNN_3 and CNN_4 (average runtime was 12 min per fold) were also faster than CNN_LSTM (average runtime was 25 min per fold). Therefore, only CNN_3 and CNN_4 were applied for further analysis.

### Effect of imbalanced data

According to the results shown in Table 4, the performance tends to decrease when the ratio of positive to negative data is increased. As shown, the models trained on the 1:1 gave

**Table 4  The performance of CNN_3 and CNN_4 on the combined dataset.** The training performance on the combined dataset with different ratios collected from 5-fold cross-validation are described.

| Site | Ratio | Model | | | | | | | |
|------|-------|-------|---|---|---|---|---|---|---|
| | | CNN_3 | | | | CNN_4 | | | |
| | | AUPRC | Precision | Recall | MCC | AUPRC | Precision | Recall | MCC |
| Donor | 1:1 | **0.987** | **0.944** | **0.969** | 0.841 | **0.988** | **0.940** | **0.973** | 0.846 |
| | 1:3 | 0.969 | 0.911 | 0.939 | **0.899** | 0.970 | 0.900 | 0.949 | **0.898** |
| | 1:5 | 0.954 | 0.892 | 0.912 | 0.882 | 0.954 | 0.875 | 0.931 | 0.882 |
| | 1:7 | 0.940 | 0.878 | 0.886 | 0.865 | 0.940 | 0.854 | 0.915 | 0.866 |
| Acceptor | 1:1 | **0.972** | **0.902** | **0.949** | **0.846** | **0.973** | **0.898** | **0.954** | **0.847** |
| | 1:3 | 0.933 | 0.850 | 0.894 | 0.827 | 0.935 | 0.839 | 0.907 | 0.828 |
| | 1:5 | 0.903 | 0.820 | 0.851 | 0.802 | 0.904 | 0.806 | 0.870 | 0.806 |
| | 1:7 | 0.877 | 0.804 | 0.811 | 0.780 | 0.878 | 0.788 | 0.835 | 0.783 |

**Notes.**
Bold styling emphasizes the highest values regarding the evaluation metrics used in the study.

**Table 5  The testing results of CNN_3 and CNN_4 performed on the combined dataset.** The testing results performed on the hold-out data.

| Site | Ratio | Model | | | | | | | |
|------|-------|-------|---|---|---|---|---|---|---|
| | | CNN_3 | | | | CNN_4 | | | |
| | | AUPRC | Precision | Recall | MCC | AUPRC | Precision | Recall | MCC |
| Donor | 1:1 | 0.801 | 0.394 | **0.963** | 0.600 | 0.802 | 0.387 | **0.964** | 0.595 |
| | 1:3 | 0.847 | 0.619 | 0.917 | 0.741 | 0.848 | 0.566 | 0.940 | 0.716 |
| | 1:5 | 0.887 | 0.758 | 0.885 | 0.807 | 0.887 | 0.705 | 0.921 | 0.792 |
| | 1:7 | **0.934** | **0.888** | 0.850 | **0.854** | **0.934** | **0.852** | 0.901 | **0.860** |
| Acceptor | 1:1 | 0.674 | 0.277 | **0.933** | 0.487 | 0.679 | 0.266 | **0.941** | 0.475 |
| | 1:3 | 0.734 | 0.475 | 0.871 | 0.623 | 0.738 | 0.463 | 0.881 | 0.618 |
| | 1:5 | 0.801 | 0.675 | 0.804 | 0.717 | 0.802 | 0.631 | 0.845 | 0.709 |
| | 1:7 | **0.886** | **0.850** | 0.761 | **0.776** | **0.888** | **0.828** | 0.805 | **0.788** |

**Notes.**
Bold styling emphasizes the highest values regarding the evaluation metrics used in the study.

the highest evaluation metrics, while the models trained on 1:7 gave the lowest in both donor and acceptor sites. However, the results shown in Table 5 suggest that using balanced data (1:1) for training resulted in over-fitting of the positive data where the model failed to classify negative data in the testing step. It can be concluded that validating the models on the balanced data may result in an overestimation of the performance. Between the two CNN-based models, the CNN_4 outperformed CNN_3, in both donor and acceptor sites, based on the highest AUPRC in the testing step. Recall, precision, and Matthews Correlation Coefficient (MCC) were used as tiebreakers. In addition, the comparison between CNN models and the existing HSplice tool was reported in Table 6.

## Effect of window sizes

As shown in Tables 6 and 7, the results of both training and testing show the same trend that using a sequence length of 398 nt performs better than other lengths. From Table 7, according to the AUPRC, considering donor sites, using 398nt does not cause the model

**Table 6  The testing results of CNN_3, CNN_4, and HSplice performed on the donor sites of the combined dataset.** The testing results performed on 20% of each hold-out set of the donor sites.

| Site | Evaluation Metrics | Model | Ratio | | | |
|---|---|---|---|---|---|---|
| | | | 1:1 | 1:3 | 1:5 | 1:7 |
| Donor | AUPRC | CNN_3 | 0.795 | 0.846 | 0.891 | *0.935* |
| | | CNN_4 | **0.798** | **0.847** | **0.892** | 0.934 |
| | | HSplice | 0.551 | 0.527 | 0.686 | 0.594 |
| | Precision | CNN_3 | **0.394** | **0.615** | **0.760** | *0.889* |
| | | CNN_4 | 0.387 | 0.561 | 0.708 | 0.850 |
| | | HSplice | 0.290 | 0.327 | 0.443 | 0.539 |
| | Recall | CNN_3 | 0.960 | 0.919 | 0.888 | 0.854 |
| | | CNN_4 | *0.962* | **0.940** | **0.922** | **0.903** |
| | | HSplice | 0.927 | 0.813 | 0.909 | 0.660 |
| | MCC | CNN_3 | **0.599** | **0.740** | **0.810** | 0.856 |
| | | CNN_4 | 0.594 | 0.713 | 0.795 | *0.861* |
| | | HSplice | 0.496 | 0.487 | 0.606 | 0.542 |

**Notes.**
Bold styling emphasizes the highest values regarding the evaluation metrics used in the study.

**Table 7  The training performance of CNN_4 on the GWH dataset with different window sizes.** The training performance on the GWH dataset with different window sizes were collected from 5-fold cross-validation. The average values of each evaluation matrix are shown.

| Site | Window size (nt) | Model | | | |
|---|---|---|---|---|---|
| | | CNN_4 | | | |
| | | AUPRC | Precision | Recall | MCC |
| Donor | 40 | 0.924 | 0.848 | 0.890 | 0.849 |
| | 80 | 0.933 | 0.871 | 0.894 | 0.866 |
| | 160 | 0.948 | 0.886 | 0.908 | 0.882 |
| | 240 | **0.951** | 0.888 | 0.916 | 0.888 |
| | 398 | **0.951** | **0.888** | **0.918** | **0.889** |
| Acceptor | 40 | 0.846 | 0.774 | 0.788 | 0.749 |
| | 80 | 0.890 | 0.818 | 0.834 | 0.801 |
| | 160 | 0.922 | 0.856 | 0.860 | 0.838 |
| | 240 | 0.933 | 0.880 | 0.870 | 0.857 |
| | 398 | **0.938** | **0.886** | **0.880** | **0.867** |

**Notes.**
Bold styling emphasizes the highest values regarding the evaluation metrics used in the study.

to perform significantly differently from using 160 nt and 240 nt (one-sided Welch's $t$-test $p = 0.1$ and $p = 0.39$), while using 40 nt cannot beat other window sizes (one-sided Welch's $t$-test $p = 9.4 \times 10^{-4}$, $p = 3 \times 10^{-6}$, $p = 4.8 \times 10^{-7}$, and $p = 3.7 \times 10^{-7}$, for 80 nt, 160 nt, 240 nt, and 398 nt, respectively). Similarly, for acceptor sites, using 40 nt is insufficient when compared to other window sizes (one-sided Welch's $t$-test $p = 2.8 \times 10^{-7}$, $p = 1.2 \times 10^{-9}$, $p = 1.8 \times 10^{-10}$, and $p = 2.1 \times 10^{-10}$, for 80 nt, 160 nt, 240 nt, and 398 nt, respectively).

**Table 8  The testing performance of the CNN_4 on the GWH dataset with different window sizes.** The testing results of the CNN_4 performed on the hold-out data with different window sizes.

| Site | Window size (nt) | Model | | | |
|---|---|---|---|---|---|
| | | CNN_4 | | | |
| | | AUPRC | Precision | Recall | MCC |
| Donor | 40 | 0.922 | 0.846 | 0.882 | 0.844 |
| | 80 | 0.931 | 0.868 | 0.886 | 0.859 |
| | 160 | 0.945 | 0.884 | 0.901 | 0.877 |
| | 240 | **0.950** | **0.887** | 0.910 | 0.883 |
| | 398 | **0.950** | **0.887** | **0.912** | **0.885** |
| Acceptor | 40 | 0.840 | 0.762 | 0.786 | 0.741 |
| | 80 | 0.887 | 0.813 | 0.831 | 0.796 |
| | 160 | 0.919 | 0.854 | 0.858 | 0.834 |
| | 240 | 0.932 | 0.878 | 0.867 | 0.855 |
| | 398 | **0.937** | **0.883** | **0.878** | **0.863** |

**Notes.**
Bold styling emphasizes the highest values regarding the evaluation metrics used in the study.

When it comes to testing, the model trained on 398 nt gave the best results for both donor and acceptor sites, as shown in Table 8. However, for donor sites, using 240 nt is not significantly different from using 398 nt as one-sided Welch's $t$-test $p = 0.37$.

## Variant effect prediction

As mentioned earlier, the best CNN-based model, or CNN_4 from the previous step, was used to predict the effect of variants on splicing events. Only variants on splice sites with a splicing probability of reference sequences over 80% were taken into account. There remained only 16,600 benign variants (9,361 in donor and 7,239 in acceptor) and 833 pathogenic variants (628 in donor and 205 in acceptor). The result shows that pathogenic variants significantly reduced the probability of sequences being splice sites compared to benign variants (one-sided Welch's $t$-test $p = 5.9 \times 10^{-6}$ and $p = 1.6 \times 10^{-10}$, for donor and acceptor sites, as shown in Figs. 3A and 3B, respectively). The effects of variants were further investigated in each position relative to the splice sites (Figs. 3C and 3D). It was shown that, for donor sites (see Fig. 3C), the model detected effects of variants on splicing within 5 bp around the splice site (15th-20th positions). However, for acceptor sites (see Fig. 3D), this range was extended further but still within 15 bp around the splice site (5th-36th positions).

Table 9 shows the $p$-value of a one-sided Welch's $t$-test comparing the score obtained from the CNN_4 and the ada_score from the dbscSNV database. In general, CNN_4 is able to differentiate the pathogenic and benign variants as the one-sided Welch's $t$-test $p < 0.05$ for both donor and acceptor sites. However, the ada_score yields lower $p$-values for both donor and acceptor sites meaning that the score from the dbscSNV better differentiates the benign and pathogenic variants.

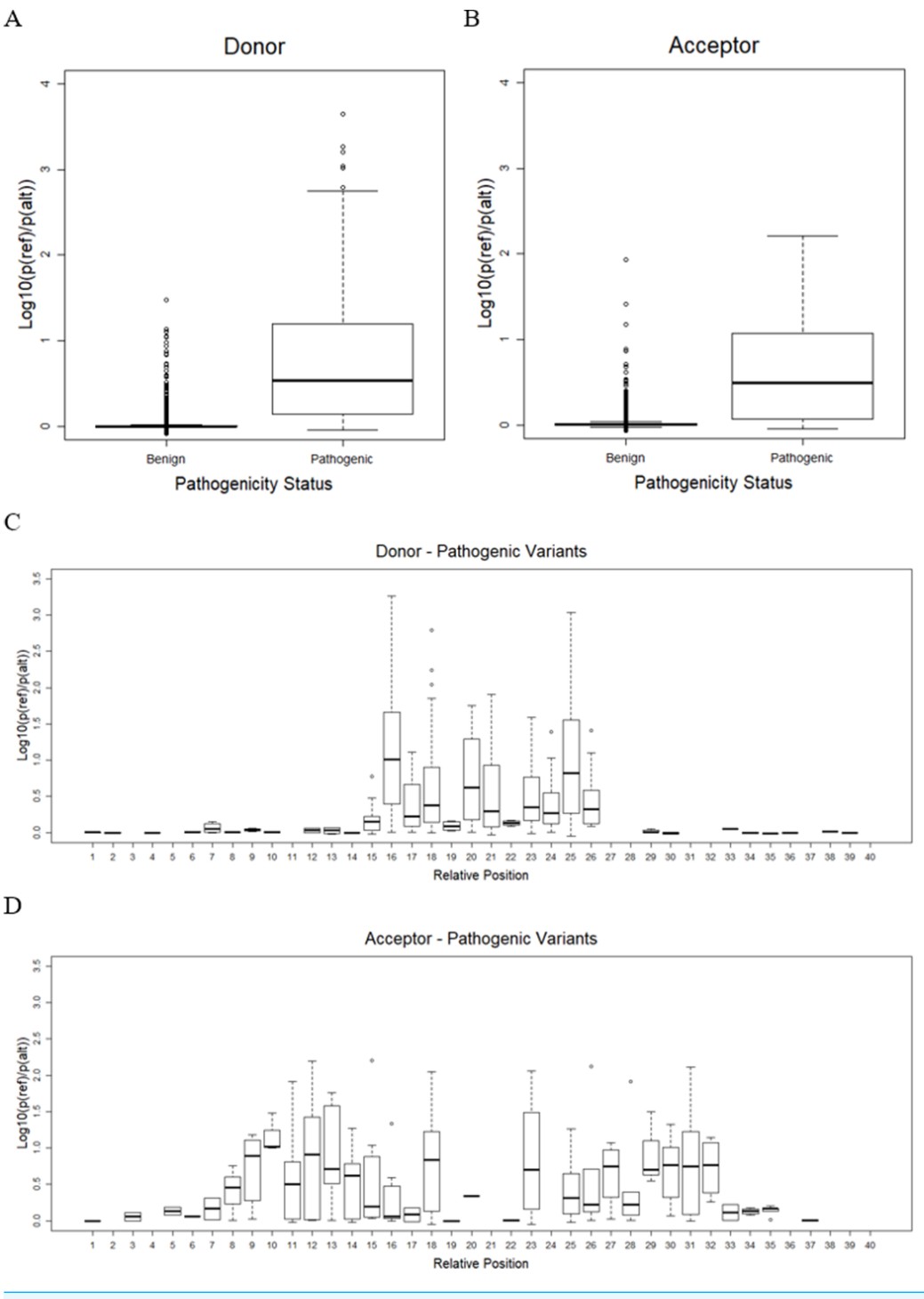

**Figure 3** **Variant effect prediction result.** (A) Distribution of donor score of benign and pathogenic variants. (B) Distribution of acceptor score of benign and pathogenic variants. (C) Distribution of donor score of variant at each position. (D) Distribution of acceptor score of variant at each position.

**Table 9  The *p*-value of one-sided Welch's *t*-test comparing between the score obtained from the CNN_4 and the ada_score from the dbscSNV database.** The *p*-value shows the comparison between the effectiveness of the CNN_4 model when predicting the effect of variants and the splicing variant scores (ada_score) from the dbscSNV database.

| Site | CNN_4 | dbscSNV |
|---|---|---|
| Donor | $1.7 \times 10^{-2}$ | $5 \times 10^{-6}$ |
| Acceptor | $3.2 \times 10^{-2}$ | $4.3 \times 10^{-3}$ |

## DISCUSSION

CNN models provided significantly better performance than the traditional machine learning approaches (see Tables 3 and 6). Based on the comparison of donor sites to the existing HSplice tool, it is clear that constructing additional features for classification using domain knowledge can provide reasonably good performance despite using only a window size of 15 nt. However, the CNN models significantly improved performance by automatically extracting more useful features than the aforementioned feature engineering process. Among CNN-based models, the models containing CNN layers alone outperformed the hybrid CNN with bi-directional LSTM. This could be because the hybrid model contains only one layer of CNN. The performance could improve or be better than CNN alone if more CNN layers are added. Even though CNN models successfully recognized the sequence patterns of the actual splice sites, there is still room for improvement. The recall of two CNN models with different complexities was very high, but their precision was not at a comparable level. As a result, the models were not as good for predicting the negative sequences. Comparing the predicting performance on the unseen data between donor and acceptor sites, the performance of acceptor site prediction was not as good as of that the donor sites. This contradicts previous studies asserting that acceptor sites are far less variable and should be easier to predict when compared to donor sites (*Garg & Green, 2007*). The limited performance in predicting negative sequences as well as acceptor sites could be caused by the limitation of the input sequence size in existing data (40 bp). In fact, more information could be fed to the model if longer sequences were used, as this study showed that the optimal performance occurs when a model is trained on input sequences of size 398 bp. However, the donor sites can perform similarly well when using 240 bp of the input sequences. For the effect of imbalanced data, according to the results shown in Table 4, training a model using balanced data (1:1) tends to cause overfitting. This may be due to the fact that the data with a 1:1 ratio does not represent the actual scenario; there is much more negative data than positive data in the human genome. It is also possible that the negative data is not enough when trying to balance the dataset by randomly using a smaller set of negative data. It seems as though the imbalanced data does not affect the model as when more negative data is added to train the model, it actually improved the performance of the model. This also applies to the traditional machine learning approach as shown in Table 6. The effectiveness of HSplice increases when the ratio of positive and negative increases. This may also be because of the fact that DL techniques, especially CNNs, can potentially extract features and learn by themselves
to address imbalanced data. However, interpretable models are still on-going in research areas.

Variant effect prediction based on the CNN model showed that the proposed framework with the CNN_4 model was able to differentiate the pathogenic splice variants from the benign splice variants. When comparing the obtained scores with the ada_score from dbscSNV database, the obtained scores for both donor and acceptor sites were not better than the ada_score in terms of the difference between benign and pathogenic variants. It could be because the dbscSNV database directly studied the variant effect on splice sites. The model was trained using variant data with given labels, while the CNN_4 model was trained by using splice sequences. Also, the dbscSNV used additional features on top of the mRNA sequences. Specifically, they used conservation scores (i.e., PhyloP46way_placental and PhyloP46way_primate) (*Jian, Boerwinkle & Liu, 2014*). Moreover, there is some overlap between their training dataset and the ClinVar database; thus, the performance observed could be due to over-fitting. Even though the CNN_4 model did not yield better results, it was able to predict a wider range of splice variants' locations, i.e., $\pm 20$ at both donor and acceptor sites, unlike the dbscSNV which can only predict $-3$ to $+8$ at donor sites and $-12$ to $+2$ at acceptor sites. Only a small set of variants could be tested here as the model was trained on a sequence of length 40 bp. Based on the prediction performance comparison of difference sequence lengths, it would be better to extend the length of splice site sequences to train the model, so that not only the performance might be improved but the effect of variants located further could also be assessed.

Future work will address the limitations of the current study. In addition, an even larger amount of data will be collected, especially negative data, to provide a more comprehensive dataset. This includes longer sequences and different preprocessing techniques, e.g., shifting the core-dinucleotides to different positions, and using two or more nucleotides for encoding the sequence instead of using one nucleotide as shown in this study, which would help to make more robust models. Also, the conservation scores can be added as an additional feature for the splice variants to improve the model performance. In addition, to facilitate other researchers who may be interested to pursue the effect of sequence length in more depth, the pseudocode of splice site and variant data preparation have been included on GitHub for open access; however, the domain knowledge is needed to finalize a validated dataset and it is also a time-consuming process.

## CONCLUSIONS

This study provided a framework for predicting the effects of variants on splice sites. A case study was demonstrated by applying the framework with two datasets. These datasets were combined to improve the power of the predictive model. Multiple measures were used to compare the performance of different models. CNN models outperformed traditional machine learning models with average AUPRC of 93% for donor sites and 88% for acceptor sites. The best model was the CNN model with four convolutional layers, which then used to analyze genetic variant data from the ClinVar database. It showed promising results in distinguishing pathogenic variants from the benign. A few limitations found in the

current study were discussed and will be further addressed in a future study. The GitHub repository for this study has been created, including the Python scripts and the datasets used in this study (see https://github.com/smiile8888/rna-splice-sites-recognition). However, the GWH dataset was not included because it has been published by *Lee & Yoon (2015*: https://dl.acm.org/doi/10.5555/3045118.3045382).

## ACKNOWLEDGEMENTS

The authors would like to thank Rena Lu, Ryan Chang, Nicola Lawford, Mark Chignell, and Dunja Matic for proofreading this work as well as Jaturong Kongmanee for proofreading the methodology section.

### Funding

This work was supported by the Petchra Pra Jom Klao Master scholarship from King Mongkut's University of Technology Thonburi, and the Canada-ASEAN Scholarships and Educational Exchanges for Development (SEED) funded by the Canadian Government. The funders had no role in study design, data collection and analysis, decision to publish, or preparation of the manuscript.

### Grant Disclosures

The following grant information was disclosed by the authors:
King Mongkut's University of Technology Thonburi.
Canada-ASEAN Scholarships and Educational Exchanges for Development (SEED).

### Competing Interests

The authors declare there are no competing interests.

### Author Contributions

- Thanyathorn Thanapattheerakul and Worrawat Engchuan conceived and designed the experiments, performed the experiments, analyzed the data, prepared figures and/or tables, authored or reviewed drafts of the paper, and approved the final draft.
- Jonathan H. Chan conceived and designed the experiments, authored or reviewed drafts of the paper, and approved the final draft.

### Data Availability

   Data and code are available at https://github.com/smiile8888/rna-splice-sites-recognition.

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
