# Peer review of "Predicting the effect of variants on splicing using Convolutional Neural Networks"

_PeerJ, doi:10.7717/peerj.9470_

## Round 0.1 · original submission · Major Revisions

· Academic Editor

Major Revisions

Two specialists in the field evaluated your manuscript. Both reviewers have concerns about your submission. Considering their evaluation, I recommend a major revision in your paper.

Reviewer 1 ·

Basic reporting

no comment

Experimental design

The conversion of sequence dataset into CNN input format should be described more clearly. For details refer my comments.

Validity of the findings

Comparative analysis with existing literature is required. Kindly refer my comments.

Additional comments

In this study, authors applied convolution neural network (CNN) for predicting splice sites as well as the effect of single nucleotide variants on splicing using benchmark data sets. This study is interesting. However, following points should be addressed before accepting for publication.

1. The input format for CNN model is of image type. Can the author explain how they convert the splice site sequences into images? To my knowledge, the binary transformation of sequences only yields a matrix. So it is necessary to know how this matrix is converted into images (kindly explain if any software is used). If there is other way of feeding the numerical data set into CNN, this should be explained in details.

2. The authors should compare the prediction accuracy with existing deep learning based methods with regard to prediction of splice sites.

3. Also, comparative analysis with existing literature should be performed in respect of predicting the effects of variants on splicing.

4. The size of the splice site data set is very large. Hence, time taken by CNN and other machine learning methods for training model must be mentioned.

5. Ready made source code of the methods should be provided for reproducibility of the work.

6. Authors should share the processed splice variants data set for reference in future study.

·

Basic reporting

Little improvement is required in the language like proper use of articles, grammatical corrections. More citations to the splice site prediction tools at least in last five years should be given with adequate discussions. Result and discussion section has to be more descriptive.

Experimental design

Research is original. The research question is well defined, relevant & meaningful but comparison with existing state of the art tools using an independent test set is missing.

Validity of the findings

Findings are valid

Additional comments

Review report
Thanapattheerakul et al., attempted to develop a statistical model for the prediction of splice sites and estimated the effect of variants on the splice motif. There work seems to be interesting and can be used in the field of genome annotation. Few revisions and concerns are to addressed before its publication in the PeerJ.
Major Concerns
There are many splice site prediction tools available in the public domain, but authors have not compared there tool with any other state of the art tools for splice site prediction using an independent test data set. Based on machine learning techniques various splice site prediction tools are available like MalDoss, DSSPred, HSplice etc. which are neither compared using an independent test data nor cited in the introduction. Though authors cited SpliceRover, they have not compared its performance with the proposed approach.
Have the authors made an comparison in terms of performance for different window sizes of splice sites with the best model before selecting the window size of 40. If not, either they should compare or refer appropriate citations based on window sizes for splice site prediction to justify their selection of 40 bp widow size.
In the model validation part, the data division for cross validation is not clear, probably the authors wanted to say 80% of the data set they have used for training and 20% for validation for each cross validation alternatively. But from there description it seems that they have used 100% of the data set for training and 20% of their training data has been used for validation each time. If the latter one is the case, then performance of their model is over estimated.
Authors should use Mathews Correlation Coefficient as an additional performance metric for the evaluation which is generally regarded as a balanced measure and can be used even if the classes are of very different sizes(imbalanced; see Boughorbel, S.B (2017)).
Overall the Results and Discussion section has to be more elaborated.
Minor Concerns
The authors mainly focused on mRNAs, hence they should specifically use mRNA instead of RNA wherever necessary.
In the abstract, "Several models were trained ...." authors should specify the exact number of models they have used. They should also mention the source of existing datasets in the abstract.
The authors missed a citation in the line no 164 and also missed the year of citation in the line no 197.
Line no 198; non-linearity relationship should be changed to non linear relationship.
Line no 235; the word splice site is missing after "donor and acceptor".
The authors should read the manuscript thoroughly for minor grammatical corrections throughout the manuscript.

---

## Round 0.2 · Minor Revisions

· Academic Editor

Minor Revisions

The manuscript has improved substantially after the first revision. Nevertheless, there is still a need for further corrections. One of the reviewers asked for a minor revision. I agree with this evaluation.

·

Basic reporting

The authors responded to most of my concerns properly. However, some more clarification/revisions are needed before its publication.

Experimental design

Well defined

Validity of the findings

Findings are valid

Additional comments

I am happy that the authors responded to most of my concerns properly. However, some more clarification/revisions are needed before its publication.
The authors mentioned that they could not compare the performance of their model with the suggested tools. As they have reported the accuracies for donor and acceptor splice sites separately, they could have compared the performance of the model with the existing tools in terms of donor splice sites separately.
The effect of the performance of the model based on different window sizes is concerned, in spite of lower accuracy with the window size 40 the authors have considered this window size for predicting the effect of variants. The prediction of the variant effect is dependent on the accuracy of the prediction of splice sites in the previous stage. Thus, my concern is won’t reliability of the prediction of variant effect be reduced with a low accuracy window size? Further, I can observe the window size of 240 has an equivalent accuracy with the window size of 398. Thus, this window size could have been explored for the prediction of the effect of variants.

Minor comments

Sonnenberg et al. proposed the SVM with weighted degree kernel to recognize splice sites (Sonnenburg et al., 2007). Meher et al. proposed SVM- and RF-based tools, named HSplice (Meher et al., 2016) and MaLDoSS (Meher, Sahu & Rao, 2016), to predict splice sites in many species, including Homo sapiens, Bos taurus, Danio rario and Caenorhabditis elegans; however, both tools can predict only donor sites.

Can be rewritten as

Sonnenberg et al. (2007) proposed the SVM with weighted degree kernel to recognize splice sites. Various SVM- and RF-based tools like HSplice (Meher et al., 2016) and MaLDoSS (Meher, Sahu & Rao, 2016), are available in the public domain for the prediction of donor splice sites in many species, including Homo sapiens, Bos taurus, Danio rario and Caenorhabditis elegans etc.

---

## Round 0.3 · Minor Revisions

· Academic Editor

Minor Revisions

The previous editor of this paper is no longer able to handle it, so I will take it from here. The reviewer provided additional minor suggestions for revising your manuscript. Please address those comments (or justify why they are not needed) and resubmit again.

·

Basic reporting

no comment

Experimental design

no comment

Validity of the findings

no comment

Additional comments

Authors have responded well to all my queries, however still some minor corrections are required.I suggest the senior authors to re-read the manuscript once again for spellings, definitions and grammar. Some of the minor comments are mentioned below:

Line no 83:
Sonnenberg et al. proposed the SVM with weighted degree kernel to recognize splice sites (Sonnenburg et al., 2007).
it should be written as "Sonnenberg et al. (2007) proposed the SVM with weighted degree kernel to recognize splice sites." as per journal format. The citation (Sonnenburg et al., 2007) is not required at the end of the sentence.

Line no 84-87:
Various SVM- and RF-based tools, for example, HSplice (Meher et al., 2016) and MaLDoSS (Meher, Sahu & Rao, 2016), have been made available in the public domain for the prediction of donor splice sites in many species, including Homo sapiens, Bos taurus, Danio rerio, and Caenorhabditis elegans
The species names in last sentence must be in italics.

Methodology, Line 264
“Model Comparison... As an additional comparison of the developed CNN models to the traditional machine approach, an existing web-based tool termed HSplice (Meher et al., 2016) was used.
in the above sentence "traditional machine approach" should be change to "traditional machine learning approach"

Line 462
"......GitHub for open access; however, note that domain knowledge is needed to finalize a validated dataset and it is also a time-consuming process.”
In the above sentence "however, note that domain knowledge" should be changed to "however, the domain knowledge".

---

## Round 0.4 · accepted · Accept

· Academic Editor

Accept

Thank you for addressing the reviewer's comments. I have no additional comments.